# Ampicillin Stability in a Portable Elastomeric Infusion Pump: A Step Forward in Outpatient Parenteral Antimicrobial Therapy

**DOI:** 10.3390/pharmaceutics15082099

**Published:** 2023-08-08

**Authors:** Lorena Rodríguez-Martínez, Ana Castro-Balado, Gonzalo Hermelo-Vidal, Enrique Bandín-Vilar, Iria Varela-Rey, Francisco José Toja-Camba, Teresa Rodríguez-Jato, Ignacio Novo-Veleiro, Pablo Manuel Varela-García, Irene Zarra-Ferro, Miguel González-Barcia, Cristina Mondelo-García, Jesús Mateos, Anxo Fernández-Ferreiro

**Affiliations:** 1Clinical Pharmacology Group, Health Research Institute of Santiago de Compostela (IDIS), 15706 Santiago de Compostela, Spain; lorenamarinoalvarez@gmail.com (L.R.-M.); ana.castro.balado@gmail.com (A.C.-B.); anxo.fernandez.ferreiro@sergas.es (A.F.-F.); 2Pharmacy Department, University Clinical Hospital of Santiago de Compostela (SERGAS), 15706 Santiago de Compostela, Spain; 3Pharmacology, Pharmacy and Pharmaceutical Technology Department, Faculty of Pharmacy, University of Santiago de Compostela (USC), 15782 Santiago de Compostela, Spain; 4Home Hospitalization Unit, University Clinical Hospital of Santiago de Compostela (SERGAS), 15706 Santiago de Compostela, Spain; 5Internal Medicine Department, University Clinical Hospital of Santiago de Compostela (SERGAS), 15706 Santiago de Compostela, Spain

**Keywords:** stability, ampicillin, elastomeric pump, outpatient parenteral antimicrobial therapy

## Abstract

Outpatient parenteral antimicrobial therapy (OPAT) with continuous infusion pumps is postulated as a very promising solution to treat complicated infections, such as endocarditis or osteomyelitis, that require patients to stay in hospital during extended periods of time, thus reducing their quality of life and increasing the risk of complications. However, stability studies of drugs in elastomeric devices are scarce, which limits their use in OPAT. Therefore, we evaluated the stability of ampicillin in sodium chloride 0.9% at two different concentrations, 50 and 15 mg/mL, in an elastomeric infusion pump when stored in the refrigerator and subsequently in real-life conditions at two different temperatures, 25 and 32 °C, with and without the use of a cooling device. The 15 mg/mL ampicillin is stable for up to 72 h under refrigeration, allowing subsequent dosing at 25 °C for 24 h with and without a cooling device, but at 32 °C its concentration drops below 90% after 8 h. In contrast, 50 mg/mL ampicillin only remains stable for the first 24 h under refrigeration, and subsequent administration at room temperature is not possible, even with the use of a cooling system. Our data support that 15 mg/mL AMP is suitable for use in OPAT if the volume and rate of infusion are tailored to the dosage needs of antimicrobial treatments.

## 1. Introduction

Outpatient parenteral antimicrobial therapy (OPAT) is defined as the administration of intravenous antimicrobial therapy in a community or outpatient setting as an alternative to hospitalization [1]. Several organizations, including the Infectious Diseases Society of America (IDSA), the British Society for Antimicrobial Chemotherapy (BSAC), and the Spanish Family Hospitalization Association, have developed guidelines for the implementation of OPAT [2,3,4]. OPAT is indicated for patients who require parenteral therapy for moderate-to-severe infection but are healthy enough to initiate or continue treatment without requiring hospitalization [5]. This strategy has several advantages, such as avoiding hospitalization due to administration of parenteral antibiotics (especially in patients requiring long-term treatment), reduction in the risk of nosocomial infections, reduction in healthcare system costs, readmission avoidance, and improving quality of life and patient satisfaction compared with inpatient care [6,7,8].

As basic requirements for OPAT, a medical team with sufficient expertise and the availability of infusion devices are necessary. For instance, Hospital Home Units (HHUs) allow outpatient treatment and follow-up of difficult and complex infectious processes, whereas programmable portable pumps or elastomeric infusion pumps (EIPs) can effectively and safely infuse most antimicrobial agents [9] by continuous infusion thanks to the elastomer that exerts a pressure that favors the administration [10].

EIPs are easy to use, economical, lightweight, quiet, and require no external power source to operate, allowing complete patient mobility and facilitating their return to work and social life [10]. For the administration of drugs via these systems, it is necessary to ensure that adequate concentrations are maintained during the administration period though stability studies.

Compared to intermittent antimicrobial infusion systems, EIPs allow the administration of a constant flow of drugs and their use is especially convenient for the administration of β-lactams, whose efficacy depends on plasma drug concentrations remaining above the minimum inhibitory concentration (MIC) of the microorganism for as long as possible in the dosing interval, i.e., they are time-dependent antibiotics. Ampicillin (AMP) is a β-lactam antibiotic widely used for the treatment of infections requiring long hospital stays such as endocarditis and meningitis, among others, making it an excellent candidate for use in an OPAT setting. Nevertheless, AMP usage in OPAT is hampered by its well-known limited stability [11]. The instability of AMP aqueous solutions is explained by its chemical structure. The core moiety of β-lactams is the β-lactam ring, which is highly susceptible to hydrolysis and strongly dependent on the pH of the solution. A minimum degradation of AMP in solution has been reported at pH 5–6 [12,13], while the degradation rate increases at lower or higher pH values. Other factors that impact AMP stability are the temperature of storage and concentration, both showing an inverse correlation, i.e., stability increases at lower temperatures, while concentrated solutions are less stable than diluted ones [10].

AMP stability in infusion solutions remains unclear. The reported shelf-life of AMP reconstituted at 20–30 mg/mL is highly variable, from 3 days to up to 7 days at 5 °C and from 8 h to 3 days at room temperature (20–25 °C) [14,15,16]. Assessment of AMP stability beyond 25 °C is desirable considering that EIPs are often exposed to elevated temperatures (near to 37 °C) derived from their proximity to the body during use, the influence of clothing, and the high ambient temperature of some geographical areas, which may compromise its stability, although not always required by regulatory agencies [17,18]. Recently, one study evaluated the stability of an infusion solution for the treatment of endocarditis that combined AMP and ceftriaxone at temperatures higher than 25 °C, reporting shelf-lives of 24 h at 30 °C and <24 h at 37 °C, respectively [19]. Less-divergent results are obtained when AMP is reconstituted at 50–60 mg/mL, but these concentrated solutions exhibit a reduced shelf-life of <24 h at 25 °C and at warmer temperatures (30–31.1 °C) [11,20,21]. Therefore, the variability in the results among the few studies that have evaluated the stability of AMP in EIPs to date precludes defining its usability period.

To overcome stability challenges, several approaches can be addressed, such as adding stabilizers/buffering compounds to the solutions, pH adjustment, or lowering the temperature [21,22]. Nakamura et al. evaluated the effect of five infusion solutions on the stability of 50 mg/mL AMP, namely normal saline, acetate ringer, 5% dextrose and dextrose-electrolyte with or without potassium. None of the buffering systems were superior to normal saline and in all cases the concentration dropped below 80% after 24 h at 25 °C, insufficient for OPAT [20]. In contrast, a four-fold increase in 60 mg/mL AMP stability at 30 °C was reported by Stiles et al., where a cooling device was used to control temperature from 6 h to 24 h and in which the concentration was over 90% of the baseline [21]. Although sparse, these results suggest that temperature control using cooling systems has a greater impact on AMP stability than buffering systems.

Currently, there are insufficient data on the stability of APM in EIPs, so it is necessary to carry out rigorous stability studies to ensure its proper use. The aim of this study was to simultaneously determine for the first time the stability of AMP at 15 mg/mL and 50 mg/mL packaged in EIPs during storage at 4–8 °C, so that the stability results of the two concentrations could be directly compared. Additionally, we evaluated the effect of a cooling system on the stability of 15 mg/mL and 50 mg/mL AMP when stored in EIPs at 25 °C and 32 °C, which simulates real-life conditions. The cooling device consisted of a portable, light, and easy-to-use cooling device designed to encase a frozen ice pack to prevent temperature-dependent drug degradation by keeping the solutions at low temperatures. The physical stability of both AMP solutions was also assessed.

## 2. Materials and Methods

### 2.1. Materials

Sodium ampicillin 50 g vials were obtained from Normon (Madrid, Spain). Sodium chloride 0.9% was purchased from Baxter (Deerfield, IL, USA) and water for injection was obtained from Grifols (Barcelona, Spain). Water, acetonitrile, ammonium formate, formic acid, and methanol UHPLC-MS-grade were purchased from VWR Chemicals (Radnor, PA, USA). Elastomeric infusion pumps DOSI-FUSER 250 mL (L25915-250D1) and cooling devices were supplied by Leventon (Barcelona, Spain). The DOSI-FUSER devices are composed of a core balloon of polyisoprene (latex-free) and a di(2-ethylhexyl) phthalate (DEHP)-free vinyl polychloride (PVC) infusion line. The cooling devices consisted of a portable cooling device with a zipper closing designed to contain one EIP and one ice pack, which must be kept at −20 °C at least for 24 h before use.

### 2.2. Elaboration and Packaging of Ampicillin Sterile Solutions

AMP vials were reconstituted with 50 mL water for injection following manufacturer instructions. The reconstituted solution was diluted with NaCl 0.9% to final concentrations of 15 mg/mL and 50 mg/mL. All DOSI-FUSER 250 mL EIPs were filled using a Repeater Pump (Baxter, Deerfield, IL, USA) to a final volume of 250 mL.

All solutions were prepared in triplicate under sterile conditions in a laminar flow cabinet for each of the three conditions evaluated.

### 2.3. Ampicillin Quantification

AMP was assayed with a previously validated method [23] using an ACQUITY UPLC H-Class System (Waters; Milfors, MA, USA) with ACQUITY PDA detector (Waters; Milfors, MA, USA). The column used was ACQUITY^®^ BEH C18 column (2.1 × 50 mm, 1.7 μm, Waters) at a temperature of 30 °C with a VanGuard 2.1 × 5 mm (Waters; Milfors, MA, USA) pre-column. The separation was achieved using a gradient elution composed of ammonium formate 20 mM pH = 6.5 (mobile phase A) and methanol/acetonitrile (75:25) (mobile phase B) with the gradient applied at a constant flow of 0.35 mL/min (Table 1). The detection wavelength was 225 nm. Data were collected and processed with Empower 3 Software (2002–2019 Waters; Milfors, MA, USA) Application Manager.

At each study time, the first 1 mL of the AMP solution from EIPs was discarded, and another 2 mL was collected on ice for analytical determination. Collected samples were accurately diluted 1:100 with cold LC-MS-grade water. Injection volume was set to 1 μL and injections were performed in triplicate.

### 2.4. Re-Validation of the Chromatographic Method

The previously validated analytical method was re-validated in this study following the European Medicines Agency (EMA) recommendations [24,25] in terms of linearity, carry-over, accuracy, precision, and limits of detection and quantification using an analytical standard (Sigma Aldrich; St. Louis, MO, USA; PHR1424; Lot# LRAD0378). For linearity, three independent calibration curves were created on different days. For carry-over, blank injections were inspected after the highest concentration of AMP in the linear range (1 mg/mL) was injected five times in a row. Precision and accuracy were tested by analyzing multiple injections in the concentration range of the working sample (0.5–0.15 mg/mL).

### 2.5. Stability Study Conditions

#### 2.5.1. Extended Storage Stability Study

Initially, 15 mg/mL and 50 mg/mL AMP EIPs were stored at 4 ± 1 °C in a cold chamber with temperature controlled by probe for 72 h to assess the stability of the two solutions under storage conditions. Samples were collected at five time points for analytical determination of AMP concentration: 0 h, 24 h, 48 h, 60 h, and 72 h. Subsequently, these AMP EIPs were stored in an ICH Memmert climatic chamber (Schwabach, Germany) at 25 ± 1 °C for 24 h to simulate a real-life situation after storage (case 1). Sample collection was performed at +8 h, +12 h, +16 h, +20 h, and +24 h (Figure 1).

#### 2.5.2. Real-Life Conditions Stability Study

A second study was carried out to test whether the use of a cooling device with ice pack would provide additional stability in real-life conditions. First, AMP solutions of 15 mg/mL and 50 mg/mL packaged in EIPs were stored at 4 ± 1 °C for 24 h. Then, they were protected with cooling devices containing an ice pack previously stored at −20 °C before being introduced in the climatic chamber at 25 ± 1 °C (case 2) or 32 ± 1 °C (case 3) (Figure 1). This last temperature was chosen as it mimics the temperature of the EIP in contact with the human body.

To ensure the cooling effect, the ice pack was placed in direct contact with the EIP surface and replaced every 12 h for one day at 25/32 °C. Temperature of EIPs was continuously monitored at 5 min intervals during these two experimental conditions (48 h in total) using a temperature Elitech RC-5 logger (London, UK). The device was placed in direct contact with the surface of the DOSI-FUSER elastomeric pump so that the recorded temperature was as close to the real temperature as possible. Sample collection was performed at +8 h, +12 h, +16 h, +20 h, and +24 h.

### 2.6. Physical Stability

Physical stability was evaluated by pH measurements and visual inspection of the color and appearance of the solutions. The pH of the AMP solutions was determined using 1 mL sample withdrawn from each EIP with a Hanna HI5221 (Hanna Instruments S.L., Eibar, Spain) probe installed in a BasiC20^®^ pH meter (Crison, L’Hospitalet de Llobregat, Spain) daily calibrated with pH = 4, pH = 7, and pH = 10 solutions (Scharlab^®^, Sentmenat, Spain). Each determination was carried out in triplicate at the same time points as for the stability study.

### 2.7. Statistical Analysis

The Pharmaceutical Codex was used to establish the expiry date of the compounded formulations, which was set at a reduction ≥10% of AMP with respect to the initial concentration [26]. Changes in pH were considered unacceptable if their values exceeded the acceptance criteria for peripheral intravenous administration [27,28]. Ampicillin content between different cases was compared with the Mann–Whitney U-Test. *p*-values of 0.05 or less were considered as statistically significant.

The results of the different assays were plotted using Graph Pad Prism^®^ v.9.0.1 software (GraphPad Software, San Diego, CA, USA).

## 3. Results

### 3.1. Re-Validation of the Chromatographic Method

A narrow, symmetrical, and well-defined chromatographic peak of AMP was obtained with an elution time of 0.92 min (Figure 2). A linear calibration curve was obtained over an AMP concentration range of 0.05–1 mg/mL (R^2^ = 0.999). The limit of detection (LOD) and limit of quantitation (LOQ) were 0.005 mg/mL and 0.01 mg/mL, respectively.

Finally, the method is precise and accurate within the parameters of the EMA in the concentration range of the working sample (0.15–0.5 mg/mL) (Table 2).

### 3.2. Chemical Stability Study

#### 3.2.1. Extended Storage Stability Study

The percentage of variation in the AMP concentration over 72 h storage at 4 ± 1 °C is shown in Figure 3, both for 15 mg/mL and 50 mg/mL solutions (case 1). In the case of the 15 mg/mL solution, the concentration remained around 95% after 72 h. However, in the 50 mg/mL solution, the concentration only remained above 90% for the first 24 h.

Considering that only the 15 mg/mL AMP solution met the stability criteria of >90% at the end of the 72 h refrigerated storage (case 1), the real-life conditions stability study was performed only with these EIPs. After 24 h at 25 ± 1 °C, the concentration of 15 mg/mL AMP solution remained >90% of the original concentration throughout the study period.

#### 3.2.2. Real-Life Conditions Stability Study

Based on the results from the extended storage stability study in refrigerated conditions, a 24 h storage period at 4 ± 1 °C was studied before the real-life conditions at 25 ± 1 °C (case 2) and 32 ± 1 °C (case 3) with EIPs placed inside a cooling device. Temperature was monitored to check whether the cooling devices could keep AMP solutions in EIPs at a temperature low enough to reduce the temperature-dependent degradation of AMP.

After refrigeration, the concentration of the 50 mg/mL solution fell below 90% after the first 8 h at 25 ± 1 °C, reaching 85.3 ± 2.2% of the initial content at the end of the 24 h real-life-conditions period (Figure 4a). In contrast, the concentration of the 15 mg/mL solution remained stable (98.7 ± 2.3%) after 24 h at 25 ± 1 °C (case 2).

When EIPs with 50 mg/mL AMP solution were stored at 32 ± 1 °C in a cooling device (case 3), AMP was not stable beyond the first 8 h, as observed in the previous real-life-conditions experiment, whereas the concentration of the 15 mg/mL AMP solution was maintained >90% up to 20 h, dropping to 82.3 ± 1.3% at 24 h. The reduced AMP stability observed at 32 °C is consistent with the higher temperatures experienced by EIPs (Figure 4).

Regarding temperature conditions, a sudden rise in temperature was registered immediately after EIPs in the cooling device were stored in the climatic chamber simulating real-life conditions. The increase was 13.1 °C and 9.8 °C in the first hour of storage at 25 ± 1 °C and 32 ± 1 °C, respectively, followed by a more moderate rise (approximately 1.1 °C/h). In both conditions, the EIPs reached the maximum temperature by the time the ice pack was replaced. A greater degradation of the AMP occurred in the time frame in which the EIPs were under the highest temperatures (32–36 h and 44–48 h) (Figure 4). Conversely, lower degradation was observed after the temperature drop caused by ice pack replacement (36–44 h) (Figure 4). This effect was more evident in the real-life study at 32 ± 1 °C.

Concentrations after 8 h at 4 ± 1 °C (case 1) were compared to those registered after 24 h at 4 ± 1 °C + 24 h at 25 ± 1 °C (case 2) or 32 ± 1 °C (case 3). Comparisons were performed for both concentrations of AMP in solution, despite the concentration of 50 mg/mL being <90% after 48 h in the three cases. AMP content in 50 mg/mL solutions after the end of the real-life study of case 2 was identical to that measured after 48 h in the refrigerator (Table 3). Similarly, no differences in AMP concentration between case 2 and case 1 were observed for 15 mg/mL solutions. In contrast, 50 mg/mL and 15 mg/mL AMP solutions after the end of the real-life study of case 3 were less stable than the same solutions after 48 h storage at 4 ± 1°C.

### 3.3. Determination of pH

Variations in the pH of the AMP solutions at 15 and 50 mg/mL concentrations over time in the extended storage stability study and real-life conditions without a cooling device are depicted in Figure 5a. The pH remained stable in both concentrations, with mean values of 9.0 ± 0.1 and 8.9 ± 0.1 for 50 mg/mL and 15 mg/mL, respectively. Similarly, pH remained within the accepted range specified for parenteral administration when cooling devices were used during real-life conditions (Figure 5b), although mean pH values at 25 ± 1 °C were higher compared to those obtained at 32 ± 1 °C (AMP 50 mg/mL: pH, mean ± SD = 8.80 ± 0.10 vs. 8.31 ± 0.16; AMP 15 mg/mL: pH, mean ± SD = 8.78 ± 0.11 vs. 8.28 ± 0.14, respectively; *p* = 0.00001 for both comparisons).

All AMP solutions in EIPs remained clear and colorless and no visible particle formation was noted via visual inspection throughout the study for all solutions.

## 4. Discussion

Optimizing treatment outcomes in infectious diseases requires attention to three interrelated factors: patient, pathogen, and drug [29]. Certain infectious pathologies require long hospital stays to complete intravenous antibiotic treatment [30,31,32,33]. These prolonged stays are associated with the appearance of complications and significant healthcare costs. Therefore, OPAT would overcome these drawbacks, providing an effective and sustainable therapeutic regimen, greater autonomy, and better quality of life for patients [9,31].

The therapeutic efficacy of antibiotics depends on their ability to achieve specific pharmacokinetic/pharmacodynamic (PK/PD) exposure targets relative to the pathogen’s minimum inhibitory concentration (MIC). For β-lactams, a free drug concentration above the minimum inhibitory concentration (fT > MIC) is the pharmacodynamic parameter related to antibiotic activity, i.e., they are time-dependent antibiotics. Prolonged infusion time is a strategy used to increase the percentage of dosing intervals in which fT > MIC [34], so there is strong evidence that supports and recommends the use of β-lactams in continuous infusion, especially in critical patients and complicated or difficult-to-access infections [35]. AMP is a β-lactam commonly used to treat infections requiring long-term hospitalization, being one of the preferred treatment options in infective endocarditis caused by *Enterococcus* [36], the third causative pathogen of this disease. The increasing incidence of endocarditis together with the relevance of these bacteria and its treatment, over 6 weeks in most cases, represent a challenge for healthcare systems [37,38]. AMP has been traditionally administered in hospitalized patients by intermittent perfusions (every 4 to 6 h) due to its short half-life, to ensure the attainment of the PK/PD target as recommended by numerous scientific societies [36,39]. Similarly, guidelines for osteomyelitis and prosthetic joint infections treatment suggest the use of AMP 2 g every 4 h, but they also include the possibility of administration through continuous infusion [26,40]. In addition, the use of AMP in continuous versus intermittent infusion has recently been studied with positive results [41,42,43].

EIPs facilitate antibiotic administration at home, reducing the manipulation of vascular accesses. The main limiting factors for their use in OPAT are ensuring adequate plasma concentrations (which would require therapeutic monitoring) and verifying the stability of antibiotic preparations [35]. The stability of an antibiotic depends on its excipients, reconstitution and dilution vehicles, pH and final antibiotic concentration, as well as storage conditions (temperature, light, and time). The composition of the infusion device may also affect the breakdown of antibiotics or their absorption by the device. All these aspects are critical to ensure drug stability as they determine therapeutic efficacy, safety, and the emergence of resistant microorganisms [44]. Numerous studies have been published recently, favoring the use of this strategy [1,10,22]; however, AMP has not been routinely included in OPAT precisely because of its limited stability after more than 24 h at room temperature, especially in countries with warmer climates [45].

Our results confirm that AMP stability in aqueous solution stored in elastomeric infusion pumps is inversely proportional to its concentration and temperature. The higher concentration studied of AMP 50 mg/mL has a low stability at 2–8 °C as it only remains above the fixed 90% stability limit for 24 h. Although we have not directly evaluated stability at 25 °C or 30 °C, the extrapolation of data from the real-life conditions with the use of a cooling device (cases 2 and 3) shows a degradation of 9% and 21% over 24 h, respectively, compared to the baseline concentration after the storage period. Considering higher degradation rates when no cooling device is used, we can assume 50 mg/mL AMP is stable for less than 24 h at both temperatures, which would preclude its use in clinical practice. This is in line with previous studies reporting a limited stability of 50–60 mg/mL AMP solutions [11,20]. Conversely, the 15 mg/mL AMP solution is more stable in all experimental conditions evaluated, as the AMP concentration remained over 90% for 72 h at 4 °C, as previously reported for a 20 mg/mL solution stored in a latex reservoir of elastomeric infusion devices [14]. Other studies have reported an additional stability of low-dose AMP solutions (24–30 mg/mL) for 7 days under refrigerated conditions [15,16]. However, the final AMP concentrations after such a long period are too close to 90% with respect to the initial concentration, so its use is not recommended since it is not possible to guarantee an adequate concentration during the administration period at room temperature [15]. Therefore, our data support that of Huskey et al. [15], showing that the chemical stability of AMP is not compromised at the end of the 24 h infusion period at 25 °C after a refrigerated storage period of 72 h. Data regarding stability of diluted AMP solutions at 25 °C are more variable. At 20 mg/mL, AMP remained stable for 8 h [14], whereas at 24 mg/mL stability was maintained for 24 h [19]. Another study with 30 mg/mL AMP reported >90% concentration after 72 h [16]. Since all AMP solutions were reconstituted in normal saline (0.9% sodium chloride), these differences may be attributed to the different material of the infusion device used in each study: latex, biocompatible DEHP polymer material/polyisoprene, and silicon, respectively, with latex being the least-suitable material. In our study, 15 mg/mL AMP placed in polyisoprene EIPs was stable for 24 h at 25 °C even after 72 h at 4 °C. Our data add to the scarce information available and point to a stability period of ≥24 h at 25 °C for AMP at low concentrations.

Data on the stability of AMP in infusion devices at temperatures over 25 °C are lacking [22], despite it being demonstrated that under real-life conditions, the temperature of these devices exceeds 25 °C and can reach 30 °C or more [18], especially in warmer geographical areas [46]. We have found only three studies that have addressed the chemical stability of AMP in infusion devices at warm temperatures. Fernández-Rubio et al. monitored the chemical stability of 24 mg/mL AMP plus 8 g/L ceftriaxone in normal saline at 30 °C and 37 °C stored in the same EIPs used in our study, but with a nominal volume on 500 mL instead of 250 mL [19]. At 30 °C, AMP recovery was >90% after 24 h, while at 37 °C, AMP concentration remained within the stability limit for approximately 6 h (extrapolated data). Nakamura et al. reported approximately 6 h of stability for 50 mg/mL AMP in normal saline at 31.1 °C [20]. Similar results were observed by Stiles et al. with 60 mg/mL AMP at 30 °C when infusion devices were immediately used or were previously stored under refrigeration for 24 h [21]. These authors designed a cooling device to encase EIPs containing the antibiotic solution and thus reduce temperature-dependent drug degradation. AMP degradation was effectively reduced at 30 °C as evidenced by the four-fold increase in its shelf life, from 6 h when no cooling device was used to 24 h. In the present work, we have also tested the impact of a cooling device on AMP stability and have observed a similar protective effect towards degradation, although of minor magnitude. This variability is probably due to the different design of the devices. The element of the design that may have had a greater impact on the cooling ability of the devices is the use of two frozen gel packs in the previous study compared to the use of only one in our study. In fact, we have found that EIPs placed in the cooling device experience a sudden increase in temperature of about 10 °C in the first hour of storage at 25 °C and 30 °C after the initial refrigeration period. Unfortunately, the temperature of the infusion pumps encased in the cooling device was not monitored in the previous study and no comparisons can be made.

Another way to increase the stability of AMP in solution could be the use of buffering systems in the reconstitution process. For instance, the addition of 10 mM sodium phosphate buffer for injection into 12 mg/mL AMP in normal saline extended its stability at room temperature two-fold compared to the stability of unbuffered solutions [47]. This effect has been attributed to the decrease in pH to around 6, where AMP is more stable compared to pH 8–9 after reconstitution. Other buffering systems have been added during AMP reconstitution, but no additional stability has been reported compared to normal saline [20]. Of note, the recovery of AMP in acetate ringer solution was comparable to that observed in normal saline. Considering that the pH of acetate ringer solution is 5.9–6.2, some stability-enhancing effect would be expected as reported for sodium phosphate for injection (pH 5–6 depending on the commercial form). The use of 0.3% (*w*/*v*) citrate-buffered saline pH 7 as a diluent has proven useful to increase the stability of piperacillin/tazobactam in elastomeric devices [48], but no investigations have been performed with AMP to date. Therefore, more effort should be made to further assess the effect of pH control using buffering systems on AMP stability [22]. Additionally, it would be of great interest to evaluate whether simultaneous control of the temperature and pH of AMP solutions has a synergistic effect on its stability.

The data derived from this work support the possibility of administering up to 15 g per day of AMP using 500 mL DOSIFUSER infusers at 15 mg/mL, exchanged by the patient himself every 12 h, since changes in volume or infusion rate are not expected to have a significant impact on the stability. Following this therapeutic regimen, EIPs could be prepared and dispensed twice a week, favoring their transfer for use in OPAT for the treatment of infections such as endocarditis or osteomyelitis. In the case of endocarditis, these results are particularly relevant since they position the dual therapy of AMP and ceftriaxone for its use in OPAT against other novel and more expensive antibiotics with promising characteristics like teicoplanin or dalbavancin, whose usefulness in the outpatient management of infective endocarditis caused by *Enterococcus faecalis* is still not sufficiently justified to make a new recommendation or to implement a safe and effective outpatient treatment protocol [45,49,50,51].

Regarding the limitations of the present study, it should be noted that only the impact of lowering the temperature on AMP stability was evaluated, but not of any buffering system. In addition, we have not tested AMP stability when directly exposed to temperatures above 25 °C during the infusion period, which limits the external validation of our results. Previous results for the stability of concentrated AMP solutions at warm temperatures have clearly shown what is already known: excessive drug degradation of >10% during the first 24 h [20,21]. A recent study with 24 mg/mL AMP reported stability >90% for the initial 24 h at 30 °C, but these EIPs did not undergo a previous refrigerated storage period [19]. Likely, similar results are expected to be reported by us and others, eventually leading us to draw the same conclusion: the stability of AMP solutions at temperatures above 25 °C is not guaranteed long enough to be used in OPAT. Therefore, we considered that it would be more informative to evaluate the stability of AMP in EIPs placed in a cooling device when exposed to 32 °C and compare it with that obtained under refrigerated conditions rather than conducting another stability study with EIPs directly exposed to body-like temperatures. Exposure temperatures of ≥37 °C were not evaluated based on previous data that strongly suggested rapid and elevated AMP degradation even with the use of a cooling device [19,21]. Another limitation is that AMP degradation products have not been identified or quantified. In our study, AMP degradation did not cause remarkable pH changes during the entire study period. Two previous studies have evaluated pH changes in parallel to ampicillin stability using similar conditions (50 mg/mL AMP diluted in normal saline, refrigerated, and stored at temperatures of 30/31.1 °C) and reported discordant results. Those reported by Stiles et al. [21] were concordant with ours, while Nakamura et al. [20] observed a more marked pH variation. The explanation for these discordant results is not straightforward. Several works have aimed to characterize the degradation products of ampicillin by using different forced conditions (acidic and alkaline hydrolysis, hot, photolytic, and humid stress conditions) [52]. Acids, diketopiperazines, and dimers/trimers of AMP are some of the characterized degradation compounds. Considering that ampicillin in infusion solutions has a basic pH, forced alkaline hydrolysis could provide a more approximate idea of the possible degradation that may occur, although the degradation behavior of ampicillin under these conditions may vary from that under more mild conditions found in EIPs. However, the lack of knowledge about AMP degradation products formed in EIPs should not lead us to forget that these solutions have a sufficiently low toxicity profile to be used for the treatment of complicated infections, provided that the AMP concentration remains >90%.

## 5. Conclusions

The limited stability of ampicillin in solution at room temperature has prevented its use in OPAT so far. The results of the present study show a stability of up to 72 h under refrigeration of 15 mg/mL AMP in 250 mL EIPs, ensuring a subsequent administration at room temperature for 24 h. In contrast, 50 mg/mL AMP can only be stored at 2–8 °C for the first 24 h and administered in the subsequent 8 h in real-life conditions even when a cooling device is used, which hinders its translation to clinical practice. Our data support that the use of 15 mg/mL AMP in 500 mL EIPs for a 12 h infusion period would ensure the administration of the indicated doses in the treatment of infectious diseases.

Thus, the present study clearly positions OPAT as a new feasible option for patients suffering from infective diseases, including endocarditis or osteomyelitis, thereby shortening their hospital stay, which will result in a better quality of life, and allowing healthcare systems to make substantial economic savings.

## Figures and Tables

**Figure 1 pharmaceutics-15-02099-f001:**
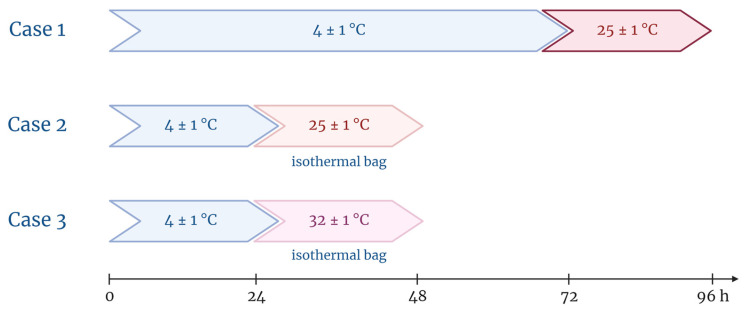
Graphical representation of the different study conditions: times, temperatures, and use of cooling device.

**Figure 2 pharmaceutics-15-02099-f002:**
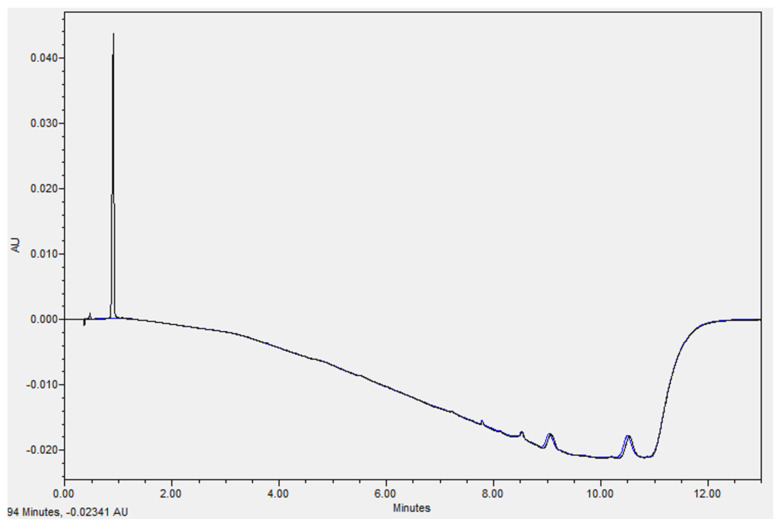
UHPLC chromatogram showing the elution profile of AMP at 225 nm (black) in the studied matrix, water (blue). Data were collected and processed with Empower^®^ 3 Software Application Manager.

**Figure 3 pharmaceutics-15-02099-f003:**
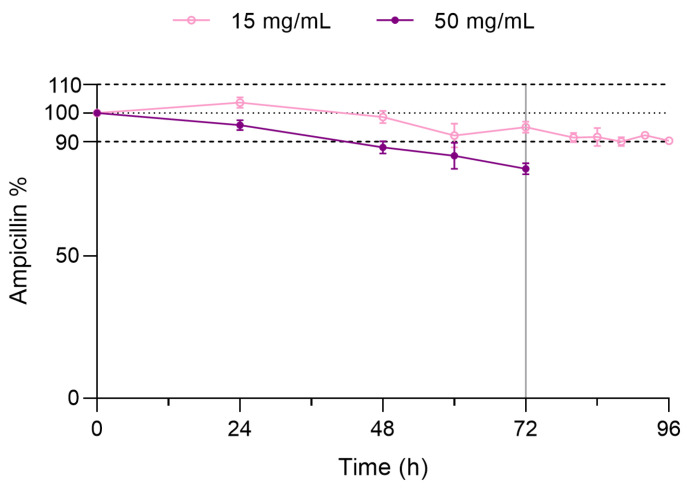
Ampicillin content expressed as percentage with respect to the initial concentration at 4 ± 1 °C for 72 h (storage period) and at 25 ± 1 °C for 24 h (real-life period). Dashed lines represent the 100% baseline concentration as well as the upper (110%) and lower limits (90%) that define the stability criteria. Vertical line at 72 h indicates the start of the real-life period at 25 °C.

**Figure 4 pharmaceutics-15-02099-f004:**
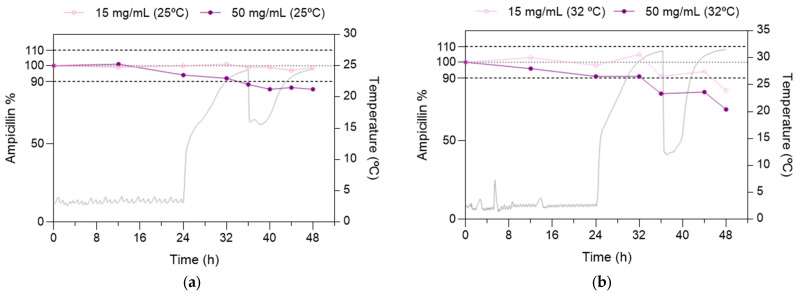
Ampicillin content expressed as percentage with respect to the initial concentration throughout the study at 4 ± 1 °C for 24 h (storage period), and subsequently simulating real-life conditions with EIPs placed inside a cooling device for the following 24 h. (**a**) at 25 ± 1 °C (case 2); (**b**) at 32 ± 1 °C (case 3). Grey line: EIPs’ temperature measures.

**Figure 5 pharmaceutics-15-02099-f005:**
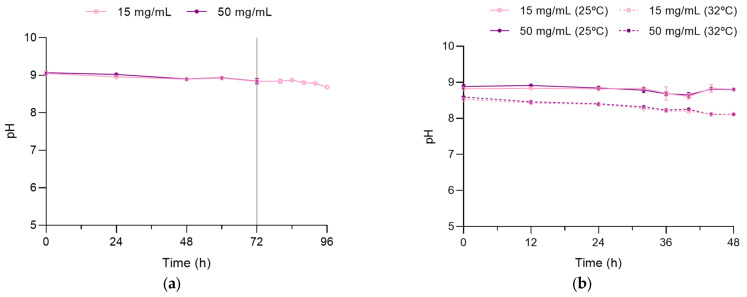
Variation in pH in 50 mg/mL and 15 mg/mL AMP solutions: (**a**) stored at 4 ± 1 °C for the first 72 h and subsequently stored for 24 h at 25 °C without cooling device (15 mg/mL) to simulate real-life conditions (case 1); (**b**) stored at 4 ± 1 °C for the first 24 h and subsequently stored for 24 h with cooling device to simulate real-life conditions at 25 ± 1 °C (case 2) or at 32 ± 1 °C (case 3). Vertical line at 72 h indicates the start of the real-life period at 25 °C.

**Table 1 pharmaceutics-15-02099-t001:** Gradient UHPLC quantification method for ampicillin. Mobile phase A was ammonium formate 20 mM pH = 6.5, and mobile phase B was methanol/acetonitrile (75:25).

Time (min)	Mobile Phase A (%)	Mobile Phase B (%)
0	75	25
2	70	30
8	35	65
10	35	65
10.01	75	25
13	75	25

**Table 2 pharmaceutics-15-02099-t002:** Precision and accuracy parameters of the chromatographic method.

Parameter	Value
Range (mg/mL)	0.05–1
Correlation coefficient (r^2^)	0.999
Carry-over	n.d.
LOD (mg/mL)	0.005
LOQ (mg/mL)	0.01
Recovery (0.5 mg/mL)	0.477
C.V. (%)	0.42

LOD = limit of detection; LOQ = limit of quantification; C.V. = coefficient of variance; n.d. = non detected

**Table 3 pharmaceutics-15-02099-t003:** Effect of the use of the cooling device described in this work on ampicillin stability during real-life conditions compared to storage in the refrigerator. Ampicillin content expressed as percentage with respect to the initial concentration (%AMP) at the end of case 2 and case 3 (after 48 h) was compared with that of case 1 after 48 h (reference) for both 50 mg/mL and 15 mg/mL solutions. *p* values < 0.05 were considered significant.

Cases	50 mg/mL %AMP	*p*	15 mg/mL %AMP	*p*
1	88.1 ± 2.1	ref	98.6 ± 2.1	ref
2	85.3 ± 2.2	0.2	98.7 ± 2.3	0.9
3	69.9 ± 5.5	0.03	82.3 ± 1.3	0.03

## Data Availability

Raw data are available to researchers upon reasonable request to co-corresponding authors.

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
