# Peer review of "Ampicillin Stability in a Portable Elastomeric Infusion Pump: A Step Forward in Outpatient Parenteral Antimicrobial Therapy"

_pharmaceutics, 2023, doi:10.3390/pharmaceutics15082099_

Round 1
Reviewer 1 Report
This is a very interesting manuscript dealing with the evaluation of chemical and physical stability of ampicillin in elastomeric devices for out side-hospital administration in treatment of serious infections like bacterial endocarditis. authors demonstrated the effect of drug concentration and temperature on degradation times. This study could be useful for hospital pharmacists associated with therapeutics in and outside hospitals. In my opinion the manuscript coudl be accepted for publication as it is.
Author Response
Thank you very much for taking your time reviewing our manuscript.

Reviewer 2 Report
This is a nice paper, well describing the problem of the preservation of antibiotics solutions at different concentrations and in different temperature conditions.
Just the Discussion is a bit too long and repetitive, some parts can be eliminated or at least concentrate. Discussions that are too long and repetitive are not read accurately, it is better to have a Dicussion with few points and very clear.
Author Response
Thank you very much for your consideration, we really appreciate it. Pertinent changes have been made so that the discussion does not become repetitive.

Reviewer 3 Report
The manuscript provides a comprehensive investigation into the stability of ampicillin in a portable elastomeric infusion pump. The authors should be commended for a thorough systemic review and meta-analysis. The following are some suggested comments, advice and corrections.
While I do not expect the authors to perform any additional stability analysis in relation to any extractables and leachables, some additional information should be provided by the authors related to published information of the type of EIP used in the study.
Ampicillin content is not the only critical quality attribute that dictates the stability of the product and any interaction between the drug product and packaging materials should at least be discussed and referenced either form information provided by the supplier / manufacturer or in other studies using the same type of product.
Line 197 - Should read Mann Whitney U-Test not U of Mann Whitney test
Update references to include all DOI values where not
The quality of English throughout the document is of a high standard and only a small number of corrections are included in the above section
Author Response
The manuscript provides a comprehensive investigation into the stability of ampicillin in a portable elastomeric infusion pump. The authors should be commended for a thorough systemic review and meta-analysis. The following are some suggested comments, advice and corrections.
Thank you for these comments, we realize the need for further details on the composition of the EIP used in our study. We hope our responses and changes have addressed this problem and satisfy your points of concern.
While I do not expect the authors to perform any additional stability analysis in relation to any extractables and leachables, some additional information should be provided by the authors related to published information of the type of EIP used in the study.
According to manufacturer’s information the Dosi-Fuser device is composed by a core balloon of polyisoprene (latex-free) and a DEHP-free vinyl polychloride (PVC) infusion line, which confers the best performance for stable fluid output and has proven drug compatibility. This information has been added to section 2.1 Materials (lines 118-120).
Ampicillin content is not the only critical quality attribute that dictates the stability of the product and any interaction between the drug product and packaging materials should at least be discussed and referenced either form information provided by the supplier / manufacturer or in other studies using the same type of product.
Thank you for raising these important considerations. We have discussed the possible influence of the EIP material on ampicillin stability by comparing the results of three studies (references 14, 16 and 19) that used similar drug concentration, diluent, and temperature (lines 351-355), so differences in stability could be attributed to the EIP material, as stated in the text (lines 352-354). According to the results, ampicillin stored in EIPs made from latex showed shorter shelf-lives compared with other materials, including the polyisoprene used here, and therefore, latex should be avoided, as indicated in the text (lines 354-355). In addition, the polyisoprene balloon of the Dosi-Fuser device has proven drug compatibility according to manufacturer’s information, as mentioned above. This information has been introduced in line 353.
Line 197 - Should read Mann Whitney U-Test not U of Mann Whitney test
We agree with your observation and the mistake was corrected.
Update references to include all DOI values where not
Thank you very much for your suggestion. DOIs have been added where possible. Changes made in the text are marked in yellow.

Reviewer 4 Report
This study aimed to investigate the stability of ampicillin in portable elastomeric infusion pump in order to be potentially used in outpatient parenteral antimicrobial therapy, which would be of great benefit to patients for several reasons (as explained in the manuscript). Overall, the study is interesting and well designed, and the manuscript is generally well written. Some new and significant results were obtained, and the conclusions comply with the objectives set.
The main drawback of this study is that it is not comprehensive. In my opinion, it should be extended with new experiments and new results (for example, the influence of pH on ampicillin stability in EIPs). Although the topic is interesting, the experimental design is very simple and the amount of results is not enough for a journal with such high impact factor in my opinion, especially considering the involvement of 14 authors in performing this study and preparing the manuscript.
Some other concerns/suggestions are as follows:
The authors stated as a limitation of that ampicillin degradation products have not been identified or quantified. It should be at least discussed from literature data, especially in the context of constant pH during the experiment despite degradation.
Line 31 (“stability drops below 90% after 8 h”) – please rephrase, it is not stability that is 90% but drug concentration as stability parameter.
Line 57 – change “his”.
Line 63 – add the abbreviation MIC here since it is the first time mentioned.
Line 79 – why did you check stability at 32 °C if it can go up to 37 °C? Please, clarify.
Line 107 – delete “+”.
Table 1 – please check the method; at minute 10 it is the same as at minute 8, so why did you write this way?
Line 212 – check the concentration range (0.05-0.15 or 0.15-0.5 mg/mL)?
Table 3 legend – please specify the end time of case 2 and case 3.
Some parts of the Discussion are too general and more appropriate for the Introduction (for example, lines 324-326).
No major issues detected.
Author Response
This study aimed to investigate the stability of ampicillin in portable elastomeric infusion pumps in order to be potentially used in outpatient parenteral antimicrobial therapy, which would be of great benefit to patients for several reasons (as explained in the manuscript). Overall, the study is interesting and well designed, and the manuscript is generally well written. Some new and significant results were obtained, and the conclusions comply with the objectives set.
Thank you for your detailed review of our work and your careful and clear comments. We share your concerns and plan to conduct future studies to address unresolved questions. Now, we provide here a response to your points of concern.
The main drawback of this study is that it is not comprehensive. In my opinion, it should be extended with new experiments and new results (for example, the influence of pH on ampicillin stability in EIPs). Although the topic is interesting, the experimental design is very simple and the amount of results is not enough for a journal with such high impact factor in my opinion, especially considering the involvement of 14 authors in performing this study and preparing the manuscript.
Our experimental design conforms to the standard for stability studies of antibiotics in EIPs, which have a clear focus on their clinical applicability. For infusion solutions, the critical factor that limits their use is drug concentration (©NHS Pharmaceutical Quality Assurance Committee, UK (NHS Pharmaceutical Quality Assurance Committee. Standard Protocol for Deriving and As-Sessment of Stability, Part 1: Aseptic Preparations (Small Molecules), 5th Ed Available online: https://www.sps.nhs.uk/wp-content/uploads/2013/12/Stability-part-1-small-molecules-5th-Ed-Sept-19.pdf) and therefore, it is the main variable studied in our study and others (references 14, 15, 16, 19, 20, 21). We already mentioned in our article that pH influences ampicillin stability. However, we decided not to study this parameter since the pH of infusion solutions is not adjusted in clinical practice. Conversely, we have evaluated the effect of lowering the temperature on the stability of ampicillin in EIP when simulating real-life conditions once the devices are delivered to patients because this strategy does not alter the preparation of the solution in clinical practice and can be easily implemented.
Some other concerns/suggestions are as follows:
The authors stated as a limitation of that ampicillin degradation products have not been identified or quantified. It should be at least discussed from literature data, especially in the context of constant pH during the experiment despite degradation.
Thank you for your interesting comment. Impurities or other degradation products were not measured as in previous publications on ampicillin stability in elastomeric infusers (references 14, 15, 16, 19, 20, 21). We acknowledge that the constant pH reported in our study despite ampicillin degradation may be striking. However, a similar observation has been reported in a previous study (reference 21) that used similar conditions: 50 mg/mL AMP diluted in normal saline, refrigerated and storage temperatures of 30ºC). Another study (reference 20) reported a more marked pH variation instead. The explanation for these discordant results is not straightforward. This information has been added to the discussion (lines 431-437).
The identification of degradation products is highly complex, requires specific equipment and experts with in-depth knowledge of the subject and experience in interpreting results to avoid incorrect or incomplete interpretations (Frański R, et al. Rapid Commun Mass Spectrom. 2014 Apr 15;28(7):713-22). Several works have aimed to characterize the degradation products of ampicillin by using different forced conditions (acidic and alkaline hydrolysis, hot, photolytic, and humid stress conditions), identifying up to 19 different compounds in a single study (Li t, et al. Rapid Commun Mass Spectrom. 2014 Sep 15;28(17):1929-36). Acids, diketopiperazines and dimers/trimers of ampicillin are some of the characterized degradation compounds. Considering that ampicillin in infusion solutions has a basic pH, forced alkaline hydrolysis could provide a more approximate idea of the possible degradation that may occur, although the degradation behavior of ampicillin under these conditions may vary from that under more mild conditions found in EIPs. However, the lack of knowledge about the degradation products of ampicillin in EIPs should not lead us to forget that these solutions have a sufficiently low toxicity profile to be used for the treatment of complicated infections if ampicillin concentration remains >90%.
We consider that a detailed discussion of ampicillin degradation compounds identified is out of the scope of this work and could unnecessary extend the discussion. In addition, due to the huge amount of data about this subject and its complexity, the review of the literature data could constitute a separate work. Therefore, a brief paragraph with some key points has been added in the discussion section (lines 437-447).
Line 31 (“stability drops below 90% after 8 h”) – please rephrase, it is not stability that is 90% but drug concentration as stability parameter.
Thank you for your suggestion. We agree and changes have been made in the manuscript and marked in yellow.
Line 57 – change “his”.
We thank you for the observation. The change has been introduced in the manuscript.
Line 63 – add the abbreviation MIC here since it is the first time mentioned.
Thank you very much for your suggestion. Abbreviation has been added.
Line 79 – why did you check stability at 32 °C if it can go up to 37 °C? Please, clarify.
We decided to check stability at 32ºC ± 1ºC following regulatory requirements for elastomeric devices, which indicate that this is the temperature that should be evaluated during the real-life conditions period. This standard is based on the rationale that, under appropriate handling, the drug solutions should not exceed this temperature throughout the infusion period. Additionally, if EIPs can be maintained at low temperatures near to 25ºC, by using insulated or cooling devices, it is appropriate to check stability at temperatures under 37ºC (©NHS Pharmaceutical Quality Assurance Committee, UK (NHS Pharmaceutical Quality Assurance Committee. Standard Protocol for Deriving and As-Sessment of Stability, Part 1: Aseptic Preparations (Small Molecules), 5th Ed Available online: https://www.sps.nhs.uk/wp-content/uploads/2013/12/Stability-part-1-small-molecules-5th-Ed-Sept-19.pdf). Previous studies have used similar temperatures to check ampicillin stability in EIPs (references 19, 20 and 21) and have reported shelf-lives of 24 hours or less, indicating that ampicillin could not be used in OPAT at temperatures higher than 30-32ºC. In fact, this has recently been confirmed by showing a shelf-life of less than 6 hours for ampicillin in EIPs at 37ºC (reference 19), clearly insufficient for OPAT. This point has been clarified in the discussion (lines 430-432).
Line 107 – delete “+”.
Thank you very much for the observation. We have corrected the mistake.
Table 1 – please check the method; at minute 10 it is the same as at minute 8, so why did you write this way?
Thank you for your comment. The percentage of the mobile phases between minute 8 and minute 10 remain constant, so these values are the same for both minutes. After minute 10, the percentages are changed to the initial conditions for 3 minutes so that in the next injection the column is stable for the reproducibility of the injections.
Line 212 – check the concentration range (0.05-0.15 or 0.15-0.5 mg/mL)?
Thank you for the observation. This mistake has been corrected in the manuscript.
Table 3 legend – please specify the end time of case 2 and case 3.
The end time of cases 2 and 3 was specified in the legend of table 3.
Some parts of the Discussion are too general and more appropriate for the Introduction (for example, lines 324-326).
Thank you for this comment. We have introduced changes in the discussion to eliminate general parts and make it less repetitive.

Round 2
Reviewer 4 Report
The authors modified the manuscript according to the suggestions raised.
I still think that this study, despite being interesting and well designed, is quite small and lacks more investigations and more results to be more comprehensive. However, I will support its publication in this form.